# Prostate Cancer Review: Genetics, Diagnosis, Treatment Options, and Alternative Approaches

**DOI:** 10.3390/molecules27175730

**Published:** 2022-09-05

**Authors:** Mamello Sekhoacha, Keamogetswe Riet, Paballo Motloung, Lemohang Gumenku, Ayodeji Adegoke, Samson Mashele

**Affiliations:** 1Department of Pharmacology, University of the Free State, Bloemfontein 9300, South Africa; 2Department of Health Sciences, Central University of Technology, Bloemfontein 9300, South Africa; 3Cancer Research and Molecular Biology Laboratories, Department of Biochemistry, College of Medicine, University of Ibadan, Ibadan 200005, Nigeria

**Keywords:** prostate cancer, prostate cancer diagnosis, genetics of prostate cancer, prostate-specific antigen (PSA), gene therapy, traditional medicine

## Abstract

**Simple Summary:**

Prostate cancer affects men of all racial and ethnic groups and leads to higher rates of mortality in those belonging to a lower socioeconomic status due to late detection of the disease. There is growing evidence that suggests the contribution of an individual’s genetic profile to prostate cancer. Currently used prostate cancer treatments have serious adverse effects; therefore, new research is focusing on alternative treatment options such as the use of genetic biomarkers for targeted gene therapy, nanotechnology for controlled targeted treatment, and further exploring medicinal plants for new anticancer agents. In this review, we describe the recent advances in prostate cancer research.

**Abstract:**

Prostate cancer is one of the malignancies that affects men and significantly contributes to increased mortality rates in men globally. Patients affected with prostate cancer present with either a localized or advanced disease. In this review, we aim to provide a holistic overview of prostate cancer, including the diagnosis of the disease, mutations leading to the onset and progression of the disease, and treatment options. Prostate cancer diagnoses include a digital rectal examination, prostate-specific antigen analysis, and prostate biopsies. Mutations in certain genes are linked to the onset, progression, and metastasis of the cancer. Treatment for localized prostate cancer encompasses active surveillance, ablative radiotherapy, and radical prostatectomy. Men who relapse or present metastatic prostate cancer receive androgen deprivation therapy (ADT), salvage radiotherapy, and chemotherapy. Currently, available treatment options are more effective when used as combination therapy; however, despite available treatment options, prostate cancer remains to be incurable. There has been ongoing research on finding and identifying other treatment approaches such as the use of traditional medicine, the application of nanotechnologies, and gene therapy to combat prostate cancer, drug resistance, as well as to reduce the adverse effects that come with current treatment options. In this article, we summarize the genes involved in prostate cancer, available treatment options, and current research on alternative treatment options.

## 1. Introduction

Prostate cancer affects middle-aged men between the ages of 45 and 60 and is the highest cause of cancer-associated mortalities in Western countries [1]. Many men with prostate cancer are diagnosed by prostate biopsy and analysis, prostate-specific antigen (PSA) testing, digital rectal examination, magnetic resonance imaging (MRI), or health screening. The risk factors related to prostate cancer include family risk, ethnicity, age, obesity, and other environmental factors. Prostate cancer is a heterogeneous disease both on the basis of epidemiology and genetics. The interplay among genetics, environmental influences, and social influences causes race-specific prostate cancer survival rate estimates to decrease, and thus, results in differences observed in the epidemiology of prostate cancer in different countries [2]. There is documented proof of a genetic contribution to prostate cancer. Hereditary prostate cancer and a genetic component predisposition to prostate cancer have been studied for years. One of the most predisposing genetic risk factors for prostate cancer is family inheritance. Twin studies and epidemiological studies have both proven the role of hereditary prostate cancer [3]. Many researchers have looked into the possible role of genetic variation in androgen biosynthesis and metabolism, as well as the role of androgens [4,5]. Genomics research has identified molecular processes that result in certain cancer developments, such as chromosomal rearrangements [2]. 

In general, gene mutations are a prevalent cause of cancer. Candidate genes for prostate cancer predisposition are genes that partake in the androgen pathway and metabolism of testosterone. The development of prostate epithelium and prostate cancer cells relies on the androgen receptor signaling pathway and testosterone [6]. The identification of cancer biomarkers and targeting of specific genetic mutations can be used for targeted treatment of prostate cancer. Biomarkers that can be used for targeted treatment are DNA tumor biomarkers, DNA biomarkers, and general biomarkers [7].

Prostate cancer can either be classified as androgen sensitive or androgen insensitive, which is an indicator of testosterone stimulation and the possible treatment option [8]. Treatment options available for prostate cancer are active surveillance, chemotherapy, radiation therapy, hormonal therapy, surgery, and cryotherapy. Treatment options delivered to a patient depend on the nature of the tumor, PSA level, grade and stage, and possible recurrence. For example, radical prostatectomy, a surgical option that involves the removal of the prostate and nearby tissues, is used in conjunction with radiation therapy for the treatment of low-risk prostate cancer [9]. For treating cancers that have spread beyond the prostate and have reoccurred, androgen-deprivation therapy, also called hormonal therapy, is recommended [1]. Each treatment is associated with severe side effects such as toxicity and reduced white and red blood cell counts, which lead to fatigue, hair loss, peripheral neuropathy, erectile incontinence and dysfunction, metastasis, and lastly, developing resistance to the initial treatment. Available treatment options are expensive and pose severe side effects. The discovery of new cost-effective chemotherapeutic agents with little or no side effects and higher efficacy is necessary [3]. In this review, we provide a holistic overview of prostate cancer, including the diagnosis of the disease, genes and mutations leading to the onset and progression of the disease, treatment options, and alternative treatment options.

## 2. Materials and Methods

In order to carry out the current review, in 2020, our team began to collect information and carry out a comprehensive search from different databases, i.e., Google Scholar, Pubmed, Springer, Elsevier ScienceDirect, and Web of Science, for studies published from 2010 to 2022, and older studies published as early as 2000 were included in this paper due to relevancy. The articles selected only utilized English texts, and searches were carried out for the following keywords and headings: ”prostate cancer”, ”prostate cancer genetics”, ”prostate cancer diagnosis and treatment”, ’ cancer statistics”, ”the prostate”, ”medicinal plants in prostate cancer treatment”, “traditional medicine”, “alternative therapy for prostate cancer”, ”nanomedicine in prostate cancer”, ”next generation sequencing”, “bioactive compounds in prostate cancer”, and ”drug repurposing in cancer”. Duplicate papers were eliminated, the data were screened, irrelevant works were factored out, and then full-text documents were screened. The inclusion criteria included several factors which involved original articles or review papers. The criteria for exclusion included articles with inadequate and irrelevant information and those without access to full text articles.

### 2.1. Epidemiology of Prostate Cancer

#### 2.1.1. Global Scale

Prostate cancer is one of the most common malignancies in men worldwide [10]. In 2018, GLOBOCAN reported approximately 1,276,106 new cases of prostate cancer resulting in about 358,989 deaths worldwide, with a higher prevalence in developed countries. On average, 190,000 new prostate cancer cases arise each year, with about 80,000 deaths occurring annually around the world [11]. The worldwide incidence of prostate cancer differs among various geographical regions and ethnic groups. Black men have the most reported incidence rates of prostate cancer in the world [12]. The incidence rates of Black Americans are approximately 60% higher than those of white men in America. The highest recorded incidence rates of prostate cancer are seen in developed countries where there is prostate cancer awareness and where prostate-specific antigen (PSA) testing is a prevalent screening practice [13]. The GLOBOCAN reports of PSA tests indicated high incidence rates in Australasia (111.6 per 100,000) and the USA (97.2 per 100,000) in the year 2012 [14]. Globally, prostate cancer is predicted to increase to approximately 1.7 million new cases and 499,000 deaths by the year 2030 because of the exponentially growing population and the large population of men who will be 65 years and older [15].

#### 2.1.2. Local Scale

Little is known about prostate cancer in African countries. Prostate cancer screening using the PSA test or digital rectal examination is not a well-established practice in Africa. There is a higher incidence rate of prostate cancer among men in Southern Africa as compared with Northern Africa [16]. In South Africa, prostate cancer is one of the most diagnosed cancers in men across the country. As recorded by the South African National Cancer Registry, the incidence rate of prostate cancer in 2007 was 29.4 per 100,000 men. In 2012, the incidence increased to 67.9 per 100,000 men [15].

### 2.2. Screening and Diagnosis of Prostate Cancer

Prostate cancer diagnoses at mature stages of the disease and failure of therapy are the main factors leading to an increased mortality rate. There is no single, specific test for prostate cancer; however, it has conventionally been diagnosed by a digital rectal examination (DRE), where a gloved finger is inserted into the patient’s rectum to assess the size of the prostate gland and any abnormalities. However, the prostate-specific antigen (PSA) test remains to be the keystone for prostate cancer screening [17]. PSA is a glycoprotein secreted by the epithelial cells of the prostate gland. It is usually found in semen, but can also be found in the bloodstream [18]. During PSA testing, blood samples are taken to test the level of PSA. Then, the blood samples are analyzed at a PSA cut-off point of 4 ng/mL. PSA levels above 4 ng/mL suggest that the patient needs further testing [19]. Patients with PSA levels between 4 ng/mL and 10 ng/mL have an approximately one in four chance of having prostate cancer. If the PSA is more than 10 ng/mL the possibility of having prostate cancer is over 50% [20]. PSA is prostate gland specific and not prostate cancer specific; therefore, prostate-specific antigen levels can indicate benign pathologies such as benign prostatic hyperplasia (BPH) and prostatitis and not prostate cancer, and men who do not have prostate cancer have also been reported to have elevated PSA levels. A prostate tissue biopsy is usually performed to confirm the presence of cancer [21].

A biopsy is a medical procedure in which a thin hollow needle is used to collect small tissue samples from the prostate gland to be observed under a microscope. The biopsy can be performed through the skin between the anus and scrotum or through the rectal wall (known as a transrectal biopsy) [22]. During a biopsy, the prostate gland is usually located with devices such as magnetic resonance imaging (MRI) and transrectal ultrasound (TRUS). An MRI scanner creates detailed images of body tissue using a strong magnetic field and radio waves [23]. MRI positive results can be used for specifically targeting abnormal areas of the prostate gland during a biopsy [24]. A multiparametric MRI can also be a triage test performed without a biopsy if the results were negative for DRE, PSA test, and MRI. A TRUS is a small probe that is deposited into the rectum of a patient. The probe emits sound waves that go through the prostate gland and produce echoes. The probe then recognizes and reads the echoes, and a computer system turns them into a black and white image of the organ [25].

Biopsy analysis is one of the most reliable methods of prostate cancer diagnosis. Tissue samples of a biopsy are studied and analyzed in the laboratory using a microscope. The cells can also be analyzed to determine how quickly cancer will spread. The biopsy results are usually reported as follows:Negative for prostate cancer, there were no cancer cells detected in the biopsy samples.Positive for prostate cancer, there were cancer cells detected in the biopsy samples.Suspicious, abnormal cells present, but may not be cancer cells [26].

However, artificial intelligence (AI) and machine learning algorithms have recently advanced, resulting in new classifications for prostate cancer. In recent years, the availability of novel molecular markers, as well as the introduction of advanced imaging techniques such as multiparametric magnetic resonance imaging (mpMRI) and prostate-specific membrane antigen positron emission tomography (PSMA-PET) scans have shifted the paradigm of prostate cancer screening, diagnosis, and treatment to a more individualized approach [27]. According to the most recent guidelines, any man at risk of prostate cancer should have an MRI of the prostate performed before obtaining a prostate biopsy [28]. This serves to minimize complications such as lower urinary tract symptoms, hematuria, and temporary erectile dysfunction. Furthermore, the number of biopsy cores obtained is linked to a higher risk of complications such as rectal bleeding, hematospermia, bleeding problems, and acute urine retention [29]. Therefore, radiomics can help with prostate volume selection and segmentation; prostate cancer (PCa) screening, detection, and classification; and risk stratification, treatment, and prognosis (Table 1).

### 2.3. Prostate Cancer and Genetics

#### Genetic Inheritance

Close family lineage is the primary risk factor for prostate cancer. Men with close relatives diagnosed with prostate cancer are at a 50% risk of developing cancer as compared with men with no family history of prostate cancer [26]. First-degree relatives with successive generations of diagnosed prostate cancer usually have early onset prostate cancer [31]. Epidemiologic studies have shown the inheritance of prostate cancer susceptibility genes. Analyses of case-control, twin, and family studies have concluded that prostate cancer risk may be a result of heritable factors. Research has shown specific gene mutations in hereditary prostate cancer and has reported that patients with these mutations have an increased risk of the disease [4]. In the genetic evaluation of inheritance, scientists use multigene sequencing of men diagnosed with prostate cancer, as well as men at high risk of developing cancer. About 5.5% of these men had detectable mutations in DNA repair genes such as *ATM*, *BRCA1*, and *BRCA2* genes. African men have certain genetic mutations that predispose them to prostate cancer; therefore, race and environmental conditions such as migration and food diets are considered to be contributing factors [21].

Cancer occurs because of changes in the DNA sequence due to mutations such as point mutations, single nucleotide polymorphisms (SNPs), and somatic copy number alterations (SCNAs) [31]. Mutations can cause prostate cells to become cancerous by turning off tumor suppressor genes and turning on oncogenes [32]. This often leads to uncontrolled cell division. Mutations in genes can be passed on from generation to generation or be acquired by an individual. Acquired mutations usually occur during DNA replication in the nucleus [33]. The common genes used as biomarkers for prostate cancer are BRCA genes, HOX genes, the ATM gene, RNase L (HPC1, lq22), MSR1 (8p), and ELAC2/HPC2 (17p11). Table 2 shows most of the genes used as biomarkers for prostate cancer.

Biomarkers show the advantages of being used for diagnostic procedures, staging, assessing the aggressiveness of the disease, and evaluating the therapeutic process. Multiple advances have been achieved through profiling technologies, including novel biomarkers that guide diagnosis and precision medicine. Modern biological markers, such as the prostate health index (PHI), the TMPRSS2-ERG fusion gene, 4K tests, and PCA3, have proven to increase PSA specificity and sensitivity, resulting in patients avoiding biopsies and reducing over diagnosis [76]. Table 3 below shows different diagnostic biomarkers and their different tests and categories. 

Figure 1 depicts the developmental stages of prostate cancer [78]. 

### 2.4. Precision Medicine for Prostate Cancer

Precision medicine is an emerging field that represents an alternative method, for some men with advanced cancer, to find gene-specific treatment for prostate cancer. It uses genetics as well as environmental biomarkers to determine diagnoses, prognosis therapeutic options for patients, and accurate dosing. Precision medicine classifies diseases using genome sequencing to identify patients who have tumors exhibiting actionable targets and promoting more informed and accurate treatment decisions [81]. Mutations in prostate cancer-related genes *BRCA1* and *BRCA2* render men with mCRPC suitable for treatment with either rucaparib or olaparib, and other prostate cancer genes that have responded well to olaparib treatment, which include *ATM*, *CDK12*, *CHECK2*, *CHECK1*, *PALB2*, *PP2R2A,* and *RAD54L* [82]. The influence of BRCA mutations on therapeutic outcomes in a study of 1302 patients with 67 BRCA mutation carriers was investigated. The results showed that patients who received prostatectomy or radiotherapy developed metastasis and had shorter survival as compared with patients who did not have mutations of the BRCA gene. This study also found that the BRCA1 gene was 12% more common than the BRCA2 gene, which was only 2% common. In a recent study, conducted in 2019, the mutation in the BRCA gene (c.4211C > G) was identified in a Chinese patient treated with radiotherapy and ADT for prostate cancer. The study indicated that prostate cancer patients with this specific mutation were sensitive to ADT as well as radiotherapy, making the treatment more effective [83]. Mutations that make it difficult to treat or design effective CRPC include the *F876L* mutation, which changes the binding ligand pocket in the AR. Similarly, the W741L/C mutation stimulates specific AR binding that is able to move AR into its active conformation. Such mutations create obstacles to designing effective treatment for CRPC [84].

### 2.5. Treatment and Management of Prostate Cancer

The prognostic factors consisting of initial PSA level, clinical TNM stage, and Gleason’s score have been considered together with other factors such as baseline urinary function, comorbidities, and age as a choice of treatment for prostate cancer [85]. Advances in prostate cancer diagnosis and treatment have enhanced clinicians’ capacities to classify patients by risk and propose therapy based on cancer prognosis and patient preference [86]. Surveillance, prostatectomy, and radiotherapy are recognized as the standard treatments for stage I–III prostate cancer patients. Androgen ablation by surgical or pharmacological castration can bring about lasting remission in all stage IV and high-risk stage III patients. In this case, first-generation antiandrogens such as flutamide and bicalutamide can aid. However, in stage IV, castration resistance, which is characterized by genomic mutations in the androgen receptor, invariably occurs, and the prognosis is poor [87]. Table 4 below summarizes prostate cancer treatment options and their adverse effects. 

#### 2.5.1. Active Surveillance

Active surveillance is a structured program that employs monitoring and expected intervention as the main techniques in the management of prostate cancer [89]. For patients who have low-risk cancers or those who have a short life expectancy, active surveillance has been recognized as the best option. The criteria for active surveillance have recommendations that are usually based on the following factors: disease characteristics, health conditions, life expectancy, side effects, and patient preference [90]. The PSA level, clinical progression, or histologic progression are used as prostate cancer trigger points [91].

The advantages of active surveillance are the preservation of erectile function, decreased costs of treatment, avoidance of needless treatment of inactive cancers, and sustaining life quality and normal activities. Its disadvantages include the likelihood of cancer metastasis before treatment, missed opportunity for a remedy, need for a complex therapy with side effects for larger and aggressive cancers, reduced chances of potency preservation mostly after surgery, chances of increased anxiety by patients, and frequent medical checks [92].

#### 2.5.2. Radical Prostatectomy

Radical prostatectomy is the procedure of medically removing the prostate gland by open and/or laparoscopic surgery [93]. The procedure requires making small incisions on the abdomen or via the perineum.

Salvage radical prostatectomy is usually recommended to patients with local recurrence in the absence of metastases after undergoing external beam radiation therapy, brachytherapy, or cryotherapy. This may, however, lead to increased morbidity. Patients younger than age 70 with organ-confined prostate cancer, with a life expectancy higher than 10 years who have little to no comorbidities, are best suited for radical prostatectomy. However, there are a few complications associated with its use. These complications include incontinence and erectile dysfunction arising from surgical damage to the urinary sphincter and erectile nerves [94].

#### 2.5.3. Cryotherapy

This method involves the use of surgical insertion of cryoprobes into the prostate under ultrasound guidance. It involves freezing of the prostate gland to a temperature from −100 °C to −200 °C for about 10 min. However, there are reports of complications associated with the use of this method, including urinary incontinence and urinary retention, erectile dysfunction, fistula, and rectal pain [95].

#### 2.5.4. Radiation

Radiation therapy is regarded as one of the most effective therapies that kills prostate cancer cells using high radiations. Radiations are sent to cancerous cells through various techniques such as brachytherapy (the use of seeds placed in the body) and external beam (where the energy is projected through the skin) to the cancerous sites. Radiation therapy aims at specifically transferring high-energy rays or particle doses directly to the prostate without affecting the normal tissues. These doses are based on the level of prostate cancer. This treatment is considered to be an acceptable therapy for patients who are not suited for surgical procedures [96]. Various techniques of radiation therapy are discussed below.

##### Brachytherapy

Brachytherapy includes the direct placement of radioactive sources into the prostate gland with the aid of seeds, injections, or wires under the guidance of transrectal ultrasound. This often involves two techniques: low dose and high dose rates. The low dose rate refers to the permanent implantation of seeds in the prostate tissue, which loses radioactivity gradually [97], and the latter refers to the supply of a dose of radiation to the prostate tissues with significant risk of leakage to other surrounding organs. The advantage associated with brachytherapy is that it can be completed within a day or less. There is a minimal risk of incontinence in patients without a previous transurethral resection of the prostate (TURP). Erectile function is also not affected. Its disadvantages are usually a requirement for general anesthesia, acute urinary retention risks, and persistent irritative voiding symptoms [98].

##### External Beam Radiation Therapy

External beam radiation therapy (EBRT) is a commonly used treatment technique that involves emitting strong X-ray beams specifically targeting the prostate tissues. It radiates higher prostate radiation doses, with less emission to the surrounding tissues. Radiation therapy is considered to be an effective intermediate-risk and high-risk prostate cancer treatment when used together with androgen deprivation therapy (ADT) [80]. It is a suitable therapy for attenuating metastasizing cancer cells. This technique is more advantageous than surgical therapy. It can treat early stages of cancer, and it is associated with fewer risks such as bleeding, myocardial infarction, pulmonary embolus, urinary incontinence, and erectile dysfunction. It can also relieve symptoms such as bone and joint pain [93]. Side effects of radiation include urinary urgency and frequency, erectile dysfunction, dysuria, diarrhea, and proctitis [97].

#### 2.5.5. Radium-223 Therapy

The radium-223 dichloride (Xofigo) technique makes use of a substance used for therapy in patients with metastatic prostate cancer that is resistant to hormone therapy. Its ability to mimic calcium makes radium-223 dichloride be selectively absorbed by the cancer cells in bone tissue. This technique has been reported to have a considerable impact on the survival and recovery of metastatic prostate cancer patients, leading to delayed onset of bone fracture and pain [85].

#### 2.5.6. Hormonal Therapy

Hormonal therapy is also known as androgen deprivation therapy (ADT). This technique is applied in the treatment of advanced and/or metastasized prostate cancer. Its therapeutic mechanism is based on the blockage of testosterone production and other male hormones, preventing them from fueling prostate cancer cells. Therefore, significantly decreased male hormonal levels are responsible for inhibition of the action of androgen on the androgen receptor [99]. This is often achieved using bilateral orchiectomy or medical castration via administration of luteinizing hormone-releasing hormone (LHRH) analogs or antagonists. LHRH analog primarily elevates the luteinizing hormone (LH) and follicle-stimulating hormone (FSH) by stimulating hypophysis receptors, thus, enabling the drug to downregulate the hypophysis receptors with concomitant reduction of LH and FSH levels, leading to suppressed testosterone production. Leuprolide, goserelin, triptorelin, and histrelin are among the common LHRH agonists. The antagonists cause action by blocking the hypophysis receptors, thereby triggering the immediate inhibition of testosterone synthesis [100]. ADT has, however, been associated with acute and long-term side effects, such as hyperlipidemia, fatigue, hot flashes, flare effect, osteoporosis, insulin resistance, cardiovascular disease, anemia, and sexual dysfunction [101].

Flutamide is a type of drugnthat is nonsteroidal and pure antiandrogenic lacking hormonal agonist activity. Flutamide is antiandrogen at the androgen-dependent accessory genitals. Its biological activity is based on 2-hydroxyflutamide. Treating prostate cancer with flutamide and an (LHRH) agonist has produced promising results. In vivo studies of flutamide have shown certain antagonist at the ventral prostate and androgen-dependent seminal vesicles [102,103]. Flutamide is known to result in hepatic dysfunction; however, a study on antiandrogen therapy (AAT) in combination with flutamide indicated that flutamide could be successful when performing regular hepatic function testing during treatment periods [104]. Maximum androgen blockade (MAB) using flutamide as a second-line hormonal therapy can give a prostate-specific antigen response without side effects, making this a possible treatment option for patients with HRPC with no bone metastases or whose cancer has progressed more than a year following first-line therapy [105].

Chlormadinone acetate (CMA) is an oral steroidal antiandrogen. Chlormadinone has proven to have anticancer activity. Similar to progesterone used in maximum androgen blockade (MAB) therapy as well as monotherapy for prostate cancer in Japan [106]. To determine the success of the antiandrogen chlormadinone acetate in treating stage A prostate cancer, a study of 111 patients who received chlormadinone acetate was conducted. The progression rates linked to antiandrogen therapy for stage A1 and A2 patients were lesser in non-treatment receiving groups, concluding that antiandrogen treatment with chlormadinone acetate inhibited the progression [107]. Chlormadinone is also used to treat benign prostatic hyperplasia, it decreases testosterone level, prostate-specific antigen (PSA) level, and prostate volume, in benign prostatic hyperplasia slowing the progression of Prostate cancer [108].

#### 2.5.7. Abiraterone

Abiraterone is a second-generation therapy targeted at adrenal and tumor androgen production. It is associated with the irreversible inhibition of the hydroxylase and lyase activities of CYP17A, AR pathways, and 3β-hydroxysteroid dehydrogenase activity, and is used to treat prostate cancer that has metastasized to other parts of the body [109]. Abiraterone has also been proven to be a potent inhibitor of other microsomal drug-metabolizing enzymes, including CYP1A2 and CYP2D6 [109]. Clinical data of abiraterone have indicated remarkable results, but there are reports of variable responses and concomitant increasing PSA levels. Abiraterone is correlated with high CYP17A upstream mineralocorticoids, with concomitant side effects including edema, hypertension, fatigue, and hypokalemia [110].

Immunotherapy or biological therapy is based on stimulating or suppressing the immune system. The treatment uses vaccines designed to work with the patient’s immune system to fight cancer cells. Sipuleucel-T (Provenge) is one of such vaccines, designed for advanced and metastatic prostate cancer cells that have developed resistance to hormone therapy. It is developed from the immune cells by collecting the white blood cells and activating them with prostatic acid phosphatase [109]. This is then associated with a protein that can trigger the immune system before infusing into the blood [99]. Sipuleucel-T (Provenge, Dendreon) is an autologous dendritic cell-based immunotherapy used in treating asymptomatic patients by assisting a patient’s immune system in fighting back cancer cells. It is intravenously administered in three doses over one month. Its lesser side effects make it more favorable compared to other chemotherapies. Its side effects include fever, nausea, chills, and muscle aches [111].

#### 2.5.8. Chemotherapy

Chemotherapy uses anticancer drugs to kill or inhibit the growth of cancerous cells. There has been progress in treatment of prostate cancer after decades of learning and understanding genetics, diagnosis, and treatment. The most common chemotherapy drug for prostate cancer is docetaxel (Taxotere) [112].

##### Docetaxel

Docetaxel is regarded as the first-line standard therapy for prostate cancer cells that are castration-resistant. It is an antimicrotubule agent which attaches to β-tubulin to inhibit microtubule depolymerization, thereby suppressing mitotic cell division and initiating apoptosis [113]. CYP3A is a major requirement for the activation of Docetaxel. The development of Docetaxel resistance has been associated with relapse. Docetaxel resistance has been attributed to increased upregulation of the multidrug resistance (MDR) 1 gene that encodes P-glycoprotein [114].

##### Cabazitaxel

Cabazitaxel is a novel antineoplastic semi-synthetic derived from the needles of various species of yew trees (Taxus). It is usually sold under the name Jevtana. Cabazitaxel is a second-generation therapy aimed at suppressing docetaxel resistance [99]. It has a low affinity for Pglycoprotein owing to its additional methyl groups. It is metabolized in the hepatic tissues by CYP3A4/5 and CYP2C8 (10–20%). Hypotension, bronchospasm, renal failure, neurotoxicity fatigue, alopecia, and generalized rash/erythema are among the common side effects associated with its use. There have also been reports of diarrheal deaths related to Cabazitaxel therapy resulting in electrolyte imbalances and dehydration [114].

##### Enzalutamide

Enzalutamide is a second-generation AR inhibitor that was recognized as one of the chemotherapeutic drugs for prostate cancer in 2012. This drug focuses on the androgen pathway and has functions such as (1) competitively inhibiting the binding of androgen to the androgen receptor, (2) inhibiting nuclear translocation and recruitment of cofactors, and (3) inhibiting the association of the activated androgen receptor. Enzalutamide targets androgens such as testosterone and dihydrotestosterone. Its therapeutic mechanism includes:Competitive inhibition of androgen binding to the androgen receptor;Inhibition of nuclear translocation and co-factor recruitment;Inhibition of the binding of DNA with activated androgen receptor.

The side effects of enzalutamide include fatigue, asthenia, diarrhea, and vomiting [115].

### 2.6. Combination Therapy

Combination therapy has been demonstrated as an effective strategy for prostate cancer treatment. Combination therapy is a strategy that was developed to treat castration-resistant prostate cancer and other forms of prostate cancer. There are no drugs to date that treat castration-resistant prostate cancer (CRPC), and currently approved treatment options either used alone or in combination therapy are useful in extending a patient’s lifespan by a few months [116]. Current treatment options used for the treatment of prostate cancer are not curative, and disease progresses to the castration-resistant phenotype over a period of time. Combination therapy with currently used treatment options for prostate cancer could successfully increase a patient’s lifespan and suppress tumors. Amongst all the available treatment strategies available for metastatic prostate cancer, androgen deprivation therapy (ADT) has more potential combination treatment compared to other therapeutic strategies for prostate cancer, and approved and currently ongoing clinical trials with ADT treatment include ADT with radiation therapy, which often treats high-risk patients to delay or prevent the disease from progressing to CRPC; (ii) ADT and chemotherapy, which in several clinical studies has shown to increase patient survival but results in adverse side effects and sometimes death; and (iii) immunotherapy and ADT, which has been reported to increase patient survival by 8.5 months [117]. Clinical trials are ongoing to analyze the effects of survival in ADT and the PSA-targeted poxviral vaccine, PROSTVAC-IF; a combination of radiation therapy with immunotherapy under ADT; a combination of chemotherapy with immunotherapy under ADT; and a combination of docetaxel under ADT [118]. There are a number of completed and ongoing clinical studies/trials for combination therapy of prostate cancer. Some of the clinical trials are listed in Table 5 and Table 6.

### 2.7. Drug Repurposing

Drug repurposing, also known as drug repositioning, reprofiling, or retasking, is a way of identifying new uses for approved drugs [119]. The advantage of drug repurposing over de novo drug development (developing new drugs) is that repurposed drug candidates have undergone extensive research in animal models and clinical trials, testing the safety, optimization, and, in most cases, formulation development of the drug, as well as pharmacokinetic and pharmacodynamic properties. This advantage usually speeds up the research and development for new use of the drug and reduces the failure rate in later efficacy testing clinical trials [120]. These previously tested drugs can rapidly progress into phase II and phase III human clinical studies, which implies that the associated drug development cost could be drastically deceased. Researchers show great interest in this phenomenon because drug repurposing alleviates the dilemma of some challenges currently faced in clinical research for finding new cancer therapies, such as drug shortage. It can take a period of 10–17 years for a development of a new drug compared to 3–12 years for repurposed drugs. Technology advances play a major role in scanning large databases and detecting key molecular similarities in different diseases to identify drugs that can be repurposed. Androgen deprivation therapy (ADT) is used to treat advanced-stage prostate cancer patients. Metformin is a drug commonly used to treat type II diabetes, repurposed to treat prostate cancer. It can be utilized to sensitize prostate cancer to the currently used standard prostate cancer therapies and improve the efficacy of treatment. It is reported that Metformin is able to increase the effectiveness of ADT for the treatment of prostate cancer [121]. Here, we discuss three main categories of drug repurposing studies for PCa, classified by different discovery and validation categories, such as the knowledge and ability of the drug to be researched. For example, ormeloxifene, a selective estrogen receptor modulator, is known for its anticancer properties in several cancers such as breast and ovarian cancers, but ormeloxifene is reported to have mediated the inhibition of oncogenic β-catenin signaling and EMT progression in prostate cancer by significantly suppressing β-catenin/TCF-4 transcriptional activity, N-cadherin, MMPs, and triggering pGSK3β expression. The other category is drugs that have been tested in assays and classified in accordance with their activity. For example, Itraconazole, an antifungal drug responsible for preventing angiogenesis and the initiation of the Hedgehog signaling pathway, was experimented in phase II clinical trials and established to be effective in patients with metastatic CRPC [122]. Table 7 shows different drugs repositioning candidates in prostate cancer clinical trial studies.

Other anticancer drugs that are currently being researched in vitro and in vivo for treatment of prostate cancer include naftopidil, an alpha blocker; niclosamide, an anti-helminthic agent; ormeloxifene, an estrogen receptor modulator; nelfinavir, an antiretroviral agent; glipizide, an antidiabetic agent; clofoctol, an antibacterial agent; and triclosan, an antibacterial agent [32]. Drug repurposing for prostate cancer presents an opportunity to address current treatment challenges. This strategy should be implemented using computational genomic and proteomic tools to assist and guide researchers in their decision making regarding patient treatment [122].

### 2.8. Treatment Challenges

Despite the various treatment options, mCRPC remains to be an incurable disease. Over time, the disease continues to develop resistance to different conventional treatment options [123]. This has led to continuous research on understanding the growth, metastasis, tumorigenesis, tumor microenvironment, and tumor environmental interactions that promote disease progression.

#### 2.8.1. Drug Resistance

Castration resistance has been reported in prostate cancer that has reached advanced stages. Castration resistance allows for androgen signaling via amplification of the androgen receptor’s synthesis of the intra-tumoral hormone, while disrupting the androgen receptor’s coexpressors and coactivators [124]. Resistance to enzalutamide and abiraterone acetate, as well as gene mutation in metastatic prostate cancer, has been attributed to the overexpression of the active androgen receptor (AR) in patients. Prostate cancer often develops owing to androgens; thus, most treatments are targeted at blocking androgen hormones. This is beneficial to anticancer drug-resistant patients.

Mutations have also been shown to contribute to drug resistance in cancer cells, allowing for bypassing of the targeted pathways. Alterations in intrinsic pathways such as the AR signaling pathways, MAPK/ERK pathway, endothelin A receptor (EAR), and Akt/PI3K pathways as well as exacerbated expression of the androgen receptor have been shown to contribute to ADT resistance [46].

#### 2.8.2. ABC Transporters

These transporters are expressed in the plasma membrane, where they serve as efflux pumps and are well-known triggers of multidrug resistance. They transport drugs and xenobiotics in and out of the cells [125]. Multidrug resistance protein (MRP) transporters MRP2, MRP3, MRP4, and MDR-1 protein (P-glycoprotein) have been reported in prostate cancer [110]. The exacerbated expression of these transporters has been implicated in the increased efflux of drugs, thereby leading to multidrug resistance. Of these transporters, MRP2 has been reported to exhibit the highest potency of resistance to natural product agents, MRP3 exhibits the lowest resistance to etoposide, and MRP4 and MRP5 are responsible for resistance to nucleoside analogs and transport cyclic nucleotides. MRP4 also influences resistance to chemotherapeutic agents such as camptothecins, cyclophosphamide, topotecan, methotrexate, and nucleoside analogs [126].

#### 2.8.3. Cytochrome P450

Cytochromes P450 are a well-known multigene superfamily of heme-containing monooxygenases that are both constitutive and inducible. They catalyze the metabolism of a variety of xenobiotics and endocrine disruptors [127]. The family including CYP2C19, CYP4B1, CYP3A5, CYP2D6, CYP1A2, and CYP1B1, has been reported in human prostate cells [128]. CYP4B1′s main functions are the metabolism and activation of arylamines via N-hydroxylation, an activity that results in bladder tumor [129]. Exacerbated expression of CYP1B1 has been implicated in the advances of drug resistance in prostate cancers. This is often achieved by 2-hydroxylation of flutamide [130]. CYP17A speeds up the process of sequential hydroxylase and the lyase steps in the androgen biosynthetic pathway in humans, thus, making it a critical therapeutic marker for prostate cancer treatment [131].

#### 2.8.4. Mutations in Androgen Receptors

Mutations in androgen receptors occur owing to a disorder in androgen sensitivity. Androgen receptor (AR) signaling plays an important role in the development, activity, and homeostasis of the prostate gland. It regulates the process of gene transcription via attaching to the androgen response elements on specific genes, as well as allowing nuclear translocation of the androgen receptor [132]. Gene changes in the AR signaling pathway (Figure 2) have been reported in prostate cancers. AR mutations were first reported in an androgen-responsive cell line, LNCap. These mutations have been implicated in the development of AR resistance arising from AR-targeted therapy [133]. This has led to the use of androgen deprivation therapy (ADT) and antihormone therapy in the treatment of advanced prostate cancer. The majority of AR mutations result in single amino acid substitutions, which are mostly found in the AR androgen-binding domain. The mutation T877A, which has been found in roughly 30% of metastatic CRPC patients, is the most common [134]. Other mutations have resulted in enhanced AR binding to coregulators, resulting in higher AR transcriptional activity vis-à-vis H874Y and W435L mutations. These mutations have been implicated in the development of AR resistance arising from AR-targeted therapy [124]. Figure 3 illustrates the transcription activity of the androgen receptor gene [135]. 

#### 2.8.5. Tumor Microenvironment

The tumor microenvironment has a crucial role in the development and progression of prostate cancer to the advanced stage, as per recent studies. According to experimental research, the milieu and malignant tumor cells have a mutually reinforcing relationship in which early changes in the microenvironment of normal tissue can foster carcinogenesis and tumor cells can foster more protumor modifications in the microenvironment [136]. A tumor microenvironment comprises a wide interlinked niche encompassing the extracellular matrix and specialized cells such as neural cells, blood vessels, immune cells, and mesenchymal/stromal stem cells, all of which secrete factors such as chemokines, cytokines, and matrix-degrading enzymes. They interact with cancer cells through paracrine and autocrine mechanisms [137,138].

According to a tumor stage-specific histological investigation, high-grade PC is linked to enhanced stromal immune cell infiltrates with a variety of cellular types [139]. Chronic stresses such as direct infection, urine reflux, a high-fat diet, and estrogens affect the prostate’s ability to become inflamed on a long-term basis [140]. The stromal compartment experiences an inflow of several immune cells, including CD3+ T-cells, macrophages, and mast cells, amid ongoing inflammation [141]. High levels of cytokines and chemokines, primary tumor necrosis factor, nuclear factor kappa B, to mention a few, are produced by inflammatory cells. The regulation of angiogenesis, cellular proliferation, and inflammation involves these proteins among others. They control the PC’s shift to the malignant phenotype [136].

The surrounding stromal agents go through complex modifications as a result of the interaction between prostatic epithelial cells and the tumor microenvironment, and these changes control the severity of the disease, its capacity to spread, and its susceptibility to traditional treatments [142,143].

### 2.9. Role of Estrogen Receptors (ERs) in Prostate Cancer Etiology and Progression

Prostate cancer is often regarded as hormone dependent, since steroid hormones direct its initiation and progression. Earlier reports have emphasized the significance of steroid levels in the etiology of PCa [144,145]. Estrogen plays an indispensable role in the secretion of male sex hormones, and it also plays cardinal roles in the growth, differentiation, and homeostasis of prostate tissues. Estrogens also contribute to the development of prostate cancer [146]. In a report from Ellem and Risbridger using aromatase knockout (KO) mice, the knockout mice could not metabolize androgens to estrogens, and it was observed that high levels of testosterone led to the development of prostate gland enlargement (prostatic hyperplasia). Meanwhile, increased estrogen and decreased testosterone levels gave rise to inflammatory events and lesions [147]. Epidemiological studies have also proposed that the serum level of estradiol and the serum estradiol/testosterone (E/T) ratio influence the initiation of PC and its progression [148]. Estrogen activities are carried out by two receptors, which are estrogen receptor α (ERα) or β (ERβ); ERα and Erβ are expressed in prostate tissue [125]. ERα is confined to the prostatic stroma, and has an indirect effect on the epithelial cells, while ERβ is found to be expressed within the epithelial domain and regulates epithelial proliferation and differentiation [149]. No less than five ERβ homologues (ERβ1, -2, -3, -4, and -5) exist in humans [145]. ERβ1 plays a functional role, while the other isoforms control its activity. The role of ERβ may consequently depend on the ratio of expression of ERβ1 and ERβ isoforms. It is known that ERα brings about the adverse effects induced by estrogens, while ERβ directs the protective and anti-apoptotic effects of estrogen in PCa [149]. On the one hand, the expression of estradiol receptor α has been found to be remarkably linked with a high Gleason’s score and poor survival rate in patients with PCa [150]; on the other hand, ERβ expression was found to be decreased or lost in the examined PCa samples [151]. Furthermore, the expression of ERβ2 and ERβ5 together has been shown to constitute a marker for biochemical relapse, post-surgery spread/metastasis, and the period to spread after radical prostatectomy in PCa patients. Based on the aforementioned, the expression of ERβ1 decreased, and that of ERβ2 and ERβ5 increased with the progression of PCa. This expression pattern corresponded with the spreading and metastasis of PCa [152]. In PCa, on the one hand, ERα has an oncogenic role and directs the deleterious effects of estrogen, which include proliferation, inflammation, and prostate carcinogenesis. Erβ, on the other hand, may elicit antitumor activity (oncosuppressor) in PCa manipulation of ERβ by ligands. Novel drug candidates might be useful in the therapeutic strategies towards PCa, specifically during the earlier stages of the disease [145].

### 2.10. Experimental Work Exploring Alternative Treatments

#### Traditional Medicine in Prostate Cancer Medicine in Prostate Cancer Treatment

Traditional medicine plays a significant role in healthcare in developing countries, and such countries also have a long history of treating different diseases and ailments. The use of medicinal plants in cancer has gained substantial attention, and recently, research is ongoing, with the National Cancer Institute (NCI) playing a pivotal role in the research of traditional medicine to treat cancer [153]. Traditional medicine is used significantly by patients with cancer to minimize side effects or used entirely as a single treatment rather than conventional therapy. This is because plants are easily accessible, effective, and affordable. Plant-derived compounds and plant extracts have been widely used due to their anti-inflammatory, antioxidant, and antimicrobial properties [154]. Various anticancer agents used in therapy today are derived from plants, for example, paclitaxel and taxol are derived from *Taxus brevifolia*, docetaxel (Taxotere) from *Taxus baccata*, and vincristine and vinblastine from *Catharanthus roseus* [155].

There is considerable proof supporting the utilization of a plant-based diet for the prohibition of acute disorders. Consumption of plant-based food provides necessary nutritional supplements and phytochemicals that aid in growth and shield against the occurrence of various acute illnesses [156]. They also offer protection against oxidative stress related to chronic disorders such as cancer. Phenolic compounds serve protective roles including antibacterial, anti-inflammatory, and anticancer roles [157]. Plants containing organosulfur compounds have chemoprotective activity. Carotenoids and polyphenols have anti-inflammatory and antioxidant activity [158]. Consequently, medicinal plants are commonly used for the treatment of cancers [159]. Several flavonoids have shown anticancer activity in the treatment of prostate cancer. Flavonoids are polyphenolic compounds characterized by a benzene ring condensed with a six-member phenyl ring attached to the carbon 2 and carbon 3 (C2 and C3) carbon positions. Among flavonoids, flavonols which can be identified by a distinctive hydroxyl group at the carbon 3 carbon position, have been reported in a number of studies, both preclinical and clinical, for their anticancer activity in prostate cancer cell lines. Flavonols, myricetin, fisetin, and kaempferol are commonly found in several fruits and vegetables and display anti-inflammatory, antiviral, antineoplastic, antibacterial, and antioxidant activity, among many others, in different cells [160].

Several specific plants have been analyzed for their activity as anticancer agents for cancer treatment. Plant anticancer activity is linked to phytochemical constituents present in extracts. Table 8 summarizes various medicinal plants used in cancer treatment [161].

### 2.11. Gene Therapy

The developments achieved in genetics, biotechnology, tumor biology, and immunology have facilitated new advancements in gene therapy. Gene therapy is a therapy that includes inserting or deleting a DNA sequence or base pair to rectify a genetic defect in a specific protein or to target a certain molecular pathway. A few gene editing technologies are currently being developed for gene therapy. Gene therapies usually involve the encapsulation of DNA nucleotides into viral and non-viral vectors that deliver the gene to a specific site, then, inserting the gene into the human genome to edit the DNA sequence and regulate cellular processes [162]. The main idea of gene therapy is to deliver exogenous nucleotides to specific DNA parts in the cells of various tissues. Viruses are well known for being efficient in transferring their genome into a host to infect it. The viral vector can be administered intravenously by injecting it directly into the targeted tissue. Non-viral vectors such as nanoparticles and polymers have also been studied for their use in gene therapy for the treatment of prostate cancer. These non-viral vectors usually condense DNA through electrostatic interactions, which also protects the genetic material from degrading. Gene therapies also explore the use of apoptosis. Failure of cells to undergo apoptosis can lead to uncontrolled cell division, which then leads to the development of cancer [163]. The suppression of apoptosis usually occurs as a result of the genetic mutations in cancerous cells. Gene therapy for prostate cancer targets apoptosis cellular pathways by introducing a gene that encodes a mediator or inducer of apoptosis in defective cells encoding an inducer, mediator, or executioner of apoptosis. Apoptosis-inducing genes, such as caspases, induce cell death in cancer cells [164]. Numerous challenges such as enhancing DNA transfer efficiency to cells, as well as immune responses that interfere with gene expression lie ahead for gene therapy. However, irrespective of the difficulties, it is definite that gene therapy will be the next up-and-coming medical technique used against prostate cancer in the future. Some clinical trial studies investigating prostate cancer therapy using gene therapy include various transgenes such as p53 and herpes simplex tk [165]. Recently used prostate cancer gene therapy procedures involve rectifying abnormal gene expression, utilizing programmed cell death mechanisms and biological pathways, specifically targeting important cell functions, initiating mutant or cell lytic suicide genes, strengthening the immune system anticancer response, and connecting treatment with radiation therapy or chemotherapy [166]. Animal studies in prostate cancer gene therapy have made use of intraprostatic administration of gene therapy delivery systems. This route of administration has been found to be more effective, as most of the dose was delivered directly to the prostate. This targeted delivery allowed the administered dose to reach prostate cancer metastasis. Lactoferrin and transferrin are multifunctional proteins that can bind to iron-binding proteins that are usually overexpressed on prostate cancer cells [167]. The proteins are responsible for regulating free iron levels. High iron levels have negative side effects such as increasing the risk of bacterial infections, as well generating free radicals and promoting the conversion of oxidation states ferrous ion (Fe2+) to ferric ion (Fe3+). Various studies in animals have used transferrin and lactoferrin for active targeting of prostate cancer cells. Prostate stem cell antigen (PSCA) is a cell surface antigen that is expressed in androgen-dependent and androgen-independent prostate cancer cells; therefore, it can be used as a marker for prostate cancer. Human epidermal growth factor receptor 2 (HER2) is another ligand that can be used as a marker for targeted treatment of prostate cancer due to mutations causing overexpression of tumor cells [168]. A study conducted on prostate cancer-induced xenograft mice models indicated that the inhibition of HER2 and epidermal growth factor receptor (EGFR) by specifically targeting tumor-initiating cells could highly improve the efficacy of the chemotherapy treatment for castration-resistant prostate cancer with activated STAT3, and could prevent metastasis EGF-induced STAT3 phosphorylation, which is responsible for enabling prostate cancer metastasis [169,170]. Various gene targeting systems have experimented on immune response treatment with a DAB-Lf dendriplex encoding IL12, which has demonstrated drastic tumor reduction in the PC3 and DU145 prostate tumors. MiRNA (miR)-205, miR-455-3p, miR-23b, miR-221, miR-222, miR-30c, miR-224, and miR-505 are downregulated in patients with prostate cancer and are known to be associated with tumor suppressors in prostate cancer cells, affecting proliferation, invasion, and aerobic glycolysis. MiR-663a and miR-1225-5p are linked to the development of prostate cancer, showing potential to be used as candidate markers. The specific functions of miR-663a and miR-1225-5p in stimulating prostate cancer growth and tumor progression are unclear [171,172,173].

### 2.12. CRISPR Cas9

The clustered regularly interspaced short palindromic repeats (CRISPR)/CRISPR-associated protein 9 (Cas9) is a natural defense mechanism found in archaea and bacteria. This system is currently being extensively researched because of its simplicity and effectiveness [145]. The ability to target intraprostatic inoculation of specific gene therapy vectors is an advantage of immunotherapy-based and cytotoxic gene therapy approaches. Because changes in DNA sequences result in mutations that cause cancer, scientists have been interested in new approaches to correct such changes by manipulating DNA [174]. The clustered regularly interspaced short palindromic repeats (CRISPR)/CRISPR-associated protein 9 (Cas9) system uses single-guide RNA (sgRNA) to identify and bind to certain DNA sequences through Watson–Crick base pairing [175]. CRISPR and the CRISPR–Cas9 (CRISPR-associated 9) system have been extensively studied and have changed the study of biological systems. CRISPR allows the precise altering, inserting, or deleting of DNA nucleotides in the target DNA sequence by initiating double-strand breaks. A guide RNA binds to Cas9, leading it to a complementary DNA target sequence, where a double-strand break is inserted to repair or edit DNA nucleotides. CRISPR can also be used for detecting DNA from RNA from cancerous cells and cancer-causing viruses. CRISPR/Cas9 delivery in nanoparticle lipid-based vectors is safer to use and effective [176]. Liposomal vectors offer a wide range of advantages and modifications, giving direct control over the physico-chemical properties of the liposomal surface, and can accommodate the conjugation of targeting ligands. An antibody-targeted delivery system of lipid nanoparticles (LNPs) was initially developed and standardized for the targeted treatment with small interfering RNA (siRNA). Recently, LNPs were used in a proof-of-concept study to target disseminated ovarian cancer in mice with CRISPR/Cas9 [177]. A study by Ye et al., 2017, analyzed the function of GPRC6A in the progression of prostate cancer progression in vitro and in animal studies. The study indicated that GPRG6A was expressed in human prostate cancer cell lines, and also showed polymorphism that improved mTOR signaling. Clustered regularly interspaced short palindromic repeats (CRISPR) and CRISPR-associated protein 9 nuclease (Cas9) (CRISPR/Cas9) were used to interrupt the GPRC6A gene in the PC-3 cell line. The results indicated that editing the GPRC6A gene using CRISPR/Cas9 stopped cell proliferation and migration in vitro, and also that osteocalcin activated the ERK, AKT, and mTOR signaling pathways. It was found that the GPRC6A gene mediated the progression of prostate cancer in animal studies mainly through assessing the response to osteocalcin in human prostate cancer xenograft models with cells expressing GPRC6A gene or the CRISPR/Cas9-mediated deletion of the gene. The findings of the study supported the use of CRISPR as a potential therapeutic target [178]. The first genome-scale CRISPRi screen in metastatic PCa models indicated that kinesin family member 4A (KIF4A) and WD repeat domain 62 (WDR62) initiate aggressive PCa. Novel targets for prostate cancer are also provided by CRISPR screen in prostate-specific cell lines, also suggesting the importance of assessing the results in other cancer cells, which may lead to the discovery of biomarkers for prostate cancer therapy [179].

### 2.13. Nanotechnology

Nanotechnology is an integrative field that combines pharmacology, biomedical science, and nanotechnology. Nanoparticles have characteristics that allow drug efficacy, can easily penetrate tumors, prevent drug degradation, and can be modified to target specific tissues [170]. Nanoparticles such as liposomes, polymers, metal nanomaterials, and porous silicon nanoparticles have been highly researched for application in prostate cancer treatment and prognosis. Active targeting nanoparticles have modified surfaces with attached antibodies, affibodies, peptides, or oligosaccharides. These targeting ligands target receptor cells on cancerous cells, such as the prostate-specific membrane antigen (PSMA) receptors on prostate cancer cells [180]. There is interest in developing nanoparticles for prostate cancer therapy due to challenges faced by currently used treatments. A study conducted at Mount Sinai New York on 16 patients used gold silica nanoparticles for localized prostate cancer. The gold silica nanoparticles absorbed infrared light at a wavelength that could penetrate biological tissues. The gold nanoparticles possessed plasmon resonance that could drastically decrease side effects related to the therapy. Patients were injected intravenously with gold nanoparticles with laser ablation. The growth of the tumor was analyzed using magnetic resonance imaging after 48 and 72 h of therapy. The results showed a decrease in tumor size with no side effects. While only a few studies have progressed to clinical trials, a study on targeted and controlled release for prostate cancer therapy has recently started clinical trials, which has led to the development of the BIND-014 docetaxel encapsulated nanoprototype [155]. The results of preclinical and clinical improvements linked to liposomal drug delivery in cancer treatment suggest that liposomal encapsulation signals a positive future for the treatment of prostate cancer. Nanocarriers have been demonstrated as useful in combination therapy, as they are able to overcome differences in pharmacokinetics in chemotherapeutic agents [173]. Combining nanotechnology and other therapeutic strategies can effectively enhance and improve the effectiveness of drugs. In prostate cancer, nanotechnology is used in diagnostics and therapeutic treatment. Not only are nanoparticles effective delivery systems, but they also improve the solubility of poorly soluble drugs, and multifunctional nanoparticles display adequate specificity toward urological cancers, bladder, renal, and prostate cancer. In a study conducted by Zhang et al., the encapsulation of docetaxel and doxorubicin in nanoparticles increased the observed cytotoxicity in prostate cancer cells [180]. Another study, conducted to assess the codelivery of doxorubicin (DOX) and docetaxel (DOC) by nanocarriers for synergistic activity, suggested that both anticancer agents DOX and DOC in the nanoparticles acted synergistically and promoted the curative effect of Dox and Doc in a xenograft mouse model, which acted on androgen-dependent and androgen-independent prostate cancer cell lines [181]. A multicenter phase II open-label clinical trial consisting of 42 patients with progressing mCRPC who received abiraterone acetate and/or enzalutamide treatment studied the safety and efficacy of a docetaxel-containing nanoparticle (BIND-014) targeting prostate-specific membrane antigen (PSMA) in metastatic castration-resistant prostate cancer. Targeted delivery of docetaxel by prostate-specific membrane antigen (PSMA)-conjugated nanoparticles was found to be clinically effective, drastically reducing circulating tumor cells [182]. A modern method of heating tumors after inoculation of magnetic nanoparticles has been extensively researched in prostate cancer clinical trials. The feasibility and tolerability were evaluated with the first prototype of an alternating magnetic field applicator in a study experimenting with magnetic nanoparticle thermotherapy alone or in combination with permanent seed brachytherapy. The results reported that magnetic nanoparticle thermotherapy had been shown to be hyperthermic and effective to thermoablative temperatures and could be achieved in the prostate at low magnetic field strengths of 4–5 kA/m [183,184].

### 2.14. Next-Generation Sequencing

Recently, the development of next-generation sequencing (NGS) technologies has proven to be a substantial advancement in the documentation of unique genetic alterations that have improved our understanding of cancer cell biology [185]. Precision medicine, also known as personalized medicine, strives to produce individualized treatment plans and do away with “one-size-fits-all” approaches to therapy [186]. The development of personalized treatment was supported by NGS, which not only increased our understanding of cancer but also gave oncologists a strong tool for understanding each patient’s disease and its distinct genetic characteristics and whole-genome mutational status [187,188]. NGS can identify tumor-specific alterations with single-nucleotide resolution [189]. The NGS technologies are whole-genome, whole-exome, RNA, reduced representation bisulfite, and chromatin immunoprecipitation sequencing. The three crucial phases in NGS are library preparation and amplification, sequencing, and data analysis [190]. Even though the Sanger sequencing and PCR methods have long been used to examine tumor biomarkers, the development of NGS has made it possible to screen more genes in a single test. Predictive biomarkers have subsequently been developed to assist in selecting the right patient populations for clinical investigations. Additionally, NGS enables researchers to identify the most prevalent known variants as well as the long tail of uncommon mutations that occur in less than 1% of patients and can offer helpful data on treatment sensitivity [187].

The application of NGS in PC genomics has significantly advanced the systematic cataloging of all DNA alterations occurring in cancer [188]. The identification and production of novel long non-coding RNAs and novel gene fusions in PC have been greatly aided by the use of RNA sequencing. This has resulted in the discovery of new recurrent alterations that have been identified, which are TMPRSS2-ERG translocation, SPOP and CHD1 mutations, and chromoplexy, and also the pathways that have been previously well-established have been validated (e.g., androgen receptor overexpression and mutations; PTEN, RB1, and TP53 loss/mutations) [189,190]. DNA sequencing is now far more sensitive and scaleable due to NGS [191]. PC continues to present a significant challenge in terms of diagnosis and prognosis due to its highly diverse nature [192]. To more accurately determine the cancer’s aggressiveness, clinicopathological and radiological data should be combined with the knowledge gathered from NGS investigations [193,194]. Despite having great hopes for NGS benefits, there are a number of limitations to the method that should be taken into consideration [194]. Firstly, there are valid arguments against NGS replacing established and thoroughly supported histopathological diagnoses. Although NGS can often be utilized to identify and subtype various cancer entities, an accurate pathological examination should always come first [195]. Second, NGS from tumor biopsies only provides limited temporal and geographical resolution of the entire tumor since it can only evaluate DNA and RNA changes in a small group of tumor cells at a particular timepoint [196]. This issue can be approached from a variety of angles, including improving spatial resolution through novel techniques, single-cell sequencing, serial analysis of circulating cell-free nucleic acids or tumor cells, or pragmatically focusing on the actionability of specific targets via functional studies [197]. Third, the creation of the software tools required for the analysis and clinical interpretation of the “big data” produced by NGS to support clinical decision making is still lagging behind the hardware infrastructure that is currently in place for its calculation, management, and storage [198]. Additionally, significant bioinformatical work is required to directly compare data obtained on various NGS platforms and evaluated by various bioinformatic pipelines and algorithms. Therefore, the success of NGS and precision oncology depends greatly on efficient communication and constructive teamwork among all parties [199].

## 3. Conclusions

Prostate cancer is one of the leading causes of death in men globally, after lung disease. Commonly mutated genes, proteins, and pathways associated with an increased risk of prostate cancer development can be used as biomarkers for the disease, which provide information on the stage and cause of cancer. Biomarkers can also give specifications on the type of treatment required for cancer. There is an urgent need for effective and targeted specific treatment for prostate cancer. The current treatments available for prostate cancer are beneficial to only a few patients, and present numerous side effects that eventually affect the quality of life of most patients. Chemotherapy, radiotherapy, and hormonal treatment have adverse side effects, including drug resistance, which remains a setback to anticancer treatment. Many medicinal plants, gene therapy, and the application of nanotechnology currently in research have proven to reduce side effects as well as restore chemosensitivity in resistant tumor cells. Medicinal plant fractions and compounds, genetic material encapsulated in target-specific nanocarriers with controlled release, and targeted therapies based on cellular pathways appear to be promising alternatives for prostate cancer treatment.

## Figures and Tables

**Figure 1 molecules-27-05730-f001:**
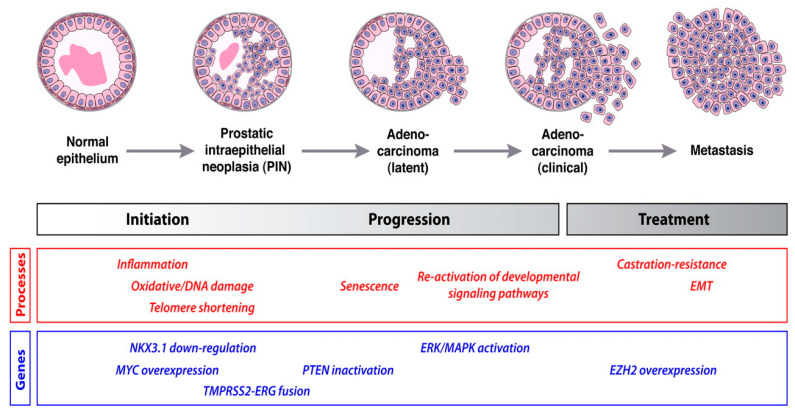
A schematic depicting the development of prostate cancer. The stages of the cancer onset and progression are indicated by the molecular processes, genes, and signaling pathways which are important in different stages of cancer. The first sign of prostate cancer is an inflammation of the prostate gland as a result of uncontrollable cell division. This uncontrollable cell division is caused by mutations that arise due to damaged DNA. At a chromosomal level, the initiation of prostate cancer begins with the shortening of telomerase at the end of the chromosome. Oxidative stress from prostate gland inflammation can shorten prostatic telomeres [78]. Research on the *Nkx3.1* homeobox gene has shown the impact of the gene on the prostate cancer initiation phase in mice. No tumor suppressor gene has been solely given a role in prostate cancer initiation or progression. However, several genes such as *MYC*, *PTEN*, *NKX3.1*., and TMPRSS2-ERG gene fusions are implicated in prostate cancer initiation. TMPRSS2-ERG gene fusions are responsible for the main molecular subtype of prostate cancer. The gene fusion activates the ERG oncogenic pathway, which contributes to the development of the disease. Metastasis of prostate cancer is conserved by the reactivation of pathways involved in cell division, which results in uncontrolled cell division and cell proliferation, leading to metastasis of the cancer [79]. Gene expression profiling results have indicated an overexpression in EZH2 mRNA and proteins present in metastatic prostate cancer. Due to the functions of EZH2 involving apoptosis and proliferation, EZH2 is a novel target for prostate cancer [80].

**Figure 2 molecules-27-05730-f002:**
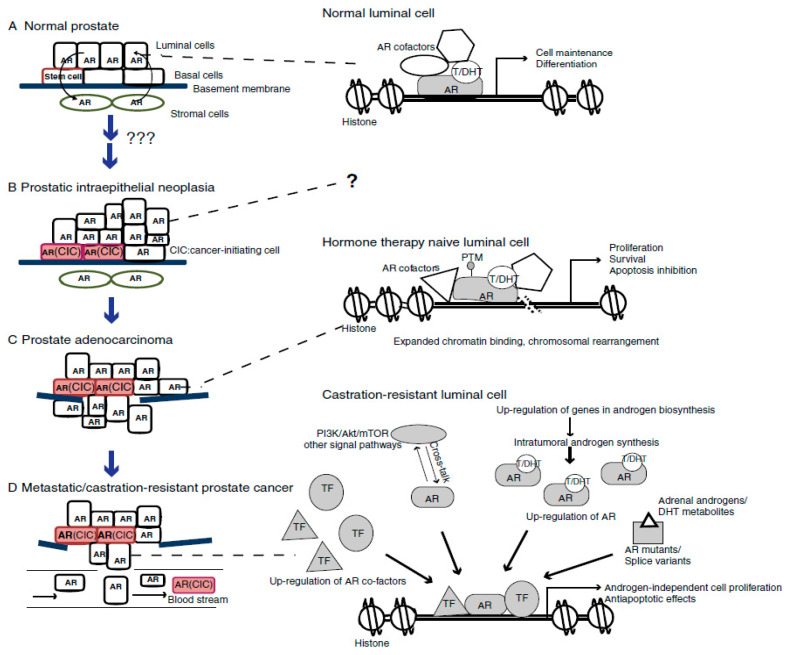
The function of AR signaling in prostate cancer and development: (**A**) Prostate homeostasis is maintained in a healthy prostate via reciprocal signaling between the stromal and epithelial layers; (**B**) normal prostate cells are converted into cancer initiating cells by unknown mechanisms, histological evidence of prostatic intraepithelial neoplasia and early cancer lesions appears, cells at the basal layer express higher levels of AR in response to this event; (**C**) cellular and molecular alterations occur in prostate adenocarcinoma, resulting in luminal cells with the AR transcriptional pathway; (**D**) Prostate cancer cells in CRPC maintain AR activity through other mechanisms (including upregulation of AR and its splice variants, intra-tumoral androgen synthesis, cross communicate with other signal pathways, and increased/altered expression of AR cofactors) as the availability of androgen from the blood steam becomes limited [134].

**Figure 3 molecules-27-05730-f003:**
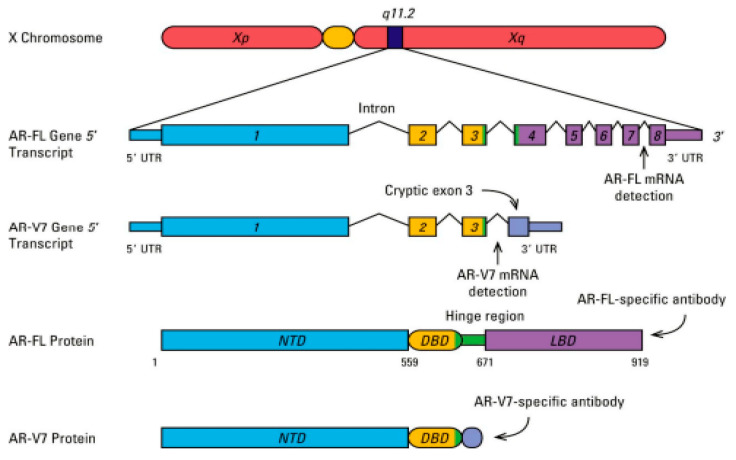
The androgen receptor gene encodes a 110 kD protein composed of 919 amino acids that are classified by an androgen-binding domain (ABD), a conserved DNA-binding domain (DBD), and an N-terminal transactivation domain, which has two polymorphic trinucleotide repeat segments. These repeated segments, consisting of variable numbers of polyglycine repeats and polyglutamine, highly influence the androgen receptor transcription activity. The gene transcript consists of eight exons in total: exon 1 codes for the N-terminal domain, exons 2–3 code for the DBD, and exons 4–8 code for the ABD [135].

**Table 1 molecules-27-05730-t001:** Benefits and drawbacks of radiogenomics as compared with actual prostate cancer peril stratification management [30].

Radiogenomics	Advantages	Limitations
	Could provide precise imaging indicators that are less expensive than genetic testing.	Lack of prospective studies
	AI and deep learning are used to produce computer-aided tools for clinical practice translation, employing large public databases containing genomes and imaging information.	Image acquisition for defining and contouring the regions of interests need expert radiologists
	Computer-designed software, both automatic and semiautomatic, is utilized to eliminate downsides (lack of standardization, imaging, and reporting protocols which differ significantly among institutions).	Significant time used for proper manual delineation
	Radiomics/radiogenomics biomarkers may be utilized to tailor treatment options and predict risk and outcomes.	Reading and segmenting regions of interest have a lot of inter-observer variability
	Biopsies are required to provide insight into the tumor genome, which is an intrusive technique that may increase patient morbidity. Tumor genetic changes can be predicted using radiogenomics.	Different acquisition techniques, scanners, and radiomic investigations, as well as a lack of repeatability and reproducibility due to a lack of standardization
	Whole-tumor data are available with a radiomics-based approach that can provide predictive and prognostic information.	Because of the differences in patient characteristics and imaging techniques, matching whole-genome sequencing data with imaging data is problematic

**Table 2 molecules-27-05730-t002:** Prostate cancer genes used as biomarkers for the disease.

Gene	Gene Description	Diagnostic/Prognostic or Predictive
*BRCA* genes	The comparative risk of prostate cancer at 65 years is 1.8–4.5-fold for *BRCA1* carriers and 2.5–8.6-fold for *BRCA2* gene carriers [31,34]. Mutations in BRCA genes inhibit DNA repair leading to prostate cancer [35].	Diagnostic Predictive [36]
*RNASEL*	Mutations in the ribonuclease L (*RNase L*) gene have been associated with prostate cancer [37]. The mutations found can inactivate the *RNase L* gene and make unsusceptible to prostate cancer [38]. RNase L is an endoribonuclease that plays a role in interferon action pathways protecting against viral infections [35,39].	Predictive [40]
*HOXB13*	*HOXB13* reduces prostate cancer growth and hormone-mediated androgen receptor activity [41,42]. The (rs339331) polymorphism increases *HOXB13* binding to a transcriptional enhancer, resulting in upregulation. Most *HOXB13* mutations correlate to the risk of aggressive and earlier-onset prostate cancer [43].	Predictive [44]
*ATM* gene	The ATM protein controls cell division and growth. It also leads to the development of certain body systems and helps cells recognize damaged DNA. Germline ATM mutations are linked to early metastasis and a lower prostate cancer survival rate [45,46].	Prognostic [47]
*HPC2* or *ELAC2* gene	*HPC2* (hereditary prostate cancer gene 2) and *ELAC2* (elaC homolog 2) are a candidate genes for hereditary prostate cancer. As with *HPC1*, mutations associated with prostate cancer are missense mutations [48].	Predictive [40]
*MSR1* gene	*MSR1* (macrophage scavenger receptor 1) at 8p22–23 of the hereditary prostate cancer (HPC) locus, and mutations linked to this gene have been associated with prostate cancer [49,50].	Predictive [40]
*ANXA7*	ANXA7 is a prostate cancer prognosis factor that shows a bimodal correlation to tumor progression [51,52]. Analyses of the ANX7 protein in prostate tumor microarrays have shown increased rates of reductions in ANX7 expression in recurrence and metastasis of hormone-refractory prostate cancer as compared with primary tumors [53].	Prognosis [51]
(*ATBF1*)-*A*	The AT-motif binding factor 1 (*ATBF1*)-*A* is a candidate for prostate cancer tumor suppression due to its function in cell inhibition and high mutation rate. A decrease in *ATBF1-A* mRNA levels is associated with a poor diagnosis. *ATBF1* inhibits cell proliferation; therefore, the loss of ATBF1 leads to uncontrolled cell growth [54,55].	Predictive [41]
*CDKN1B*	The CDKN1B’s main function is cell cycle gatekeeping. Research indicates that the CDKN1B gene is a vital tumor suppressor gene in prostate cancer. There is a correlation between the location of the CDKN1B gene (12p13) and susceptibility to prostate cancer in different populations [56,57].	Prognostic [58]
(*KLF6*) gene	Kruppel-like factor 6 (*KLF6*) is a tumor suppressor gene and a zinc finger transcription factor. In a study by Narla et al., 2008, an allele in the *KLF6* gene was deleted in 77% of prostate tumors, and the normal KLF6 gene upregulated p21 (WAF1/CIP1) and decreased cell proliferation. The KLF6-SV1 mutation overexpression elevated metastasis [59,60].	Predictive [60]
MYC gene	MYC proto-oncogene, BHLH transcription factor encodes transcription factors, promoting tumorigenesis in prostate cancer. Studies show that prostate cancer tumor foci show overexpression of MYC and protein, which is associated with the severity of the cancer. TMPRSS2-ERG gene fusion caused by a mutation of the MYC is linked to the aggressiveness of prostate cancer and seen in 60% patients [61,62,63].	Predictive [63]
*NK3* *21*	NK3 homeobox 1 (Nkx3.1) gene expression is usually lost during the process of prostate cancer initiation and growth in humans and mouse models. It was found that the loss of Nkx3.1 expression intercedes at the transcriptional stage via the 11 kb region [64,65].	Diagnostic [64]
PON1	Paraoxonase 1 (PON1) is a protein coding gene. The gene reduces oxidative stress, which leads to cancer development [66]. A study by Stevens et al., 2008, investigated the relationship between SNPs (Q192R and L55M) and prostate cancer. The results showed that the presence of a variant allele found in the Q192R and L55M SNPs was linked to an increased risk of aggressive prostate cancer [67].	Prognostic [66]
PTEN	Loss of phosphatase and tensin homolog *PTEN* is common in androgen-independent prostate cancer [68,69]. The loss of function in the *PTEN* gene is linked to irregular cellular proliferation. Studies have shown that mutations in the *PTEN* gene play a role in prostate carcinogenesis [70]. The *PTEN* gene is mutated in the prostate cell lines LNCaP, PC3, and DU145, and prostate cancer xenografts [71].	Prognostic [70]
mtDNA	Mitochondrial DNA has 16,569 bases that encode 37 genes. Mutations found in mitochondrial DNA genes have been found to cause prostate cancer [72]. In a study on mtDNA genes, approximately 12% of patients had mutations in cytochrome oxidase subunit I (COI) [73].	Prognostic [73]
RAS	Rat sarcoma virus (RAS) is part of a family of genes consisting of the N-RAS H-RAS and K-RAS, which are important in cell signaling. Point mutations that happen at codons 12, 13, or 61 of the family genes allow the protooncogene to be translated to a RAS oncogene [74].	Diagnostic [75]

**Table 3 molecules-27-05730-t003:** Examples of other diagnostic biomarkers classified as serum-based, urine-based, and tissue-based biomarkers used for prostate cancer [77].

Biomarker	Test	Category
Prostate-specific antigen	A PSA count >4 ng/mL has a specificity of 94%, but only 20% sensitivity in PCa detection; only 1 in 4 men with elevated PSA will be diagnosed with PCa.	Serum-based biomarker Standard prostate cancer screening method
4K score kallikrein markers	The 4K test includes a PCa diagnostic algorithm that includes four kallikreins in blood plasma. The analysis includes a 4K panel = total PSA (tPSA), free PSA (fPSA), intact PSA, and human kallikrein 2 (hK2).	Serum-based biomarker Detection of high-grade PCa in previously unscreened men with elevated PSA
Prostate health index (PHI)	PHI result = (−2) (proPSA/fPSA) x √ tPSA). First, the PHI test was developed to predict the probability of PCa. The use of the PHI with a cut-off ≥25 could avoid 40% of biopsies.	Serum-based biomarker Detection of any PCa PHI test also makes it possible to examine the possibility of PCa progression during active surveillance
SelectMDx HOXC6, KLK3, DLX1 mRNA, and PSAd	SelectMDx test analyzes urine samples obtained after strokes of prostate during DRE. The presence of the HOXC6 and DLX1 genes is assessed to assess the risk of any PCa during biopsy, and the risk of high-grade PCa.	Urine-based biomarker mpMRI outcomes indicate that SelectMDx score is a promising tool in PCa detection
TMPRSS2-ERG Fusion	TMPRSS2-ERG levels are linked to castration-resistant PCa. Fusion trans-membrane serine protease 2 (TMPRSS2) and ERG gene can be detected in 50% of PCa patients.	Urine-based TMPRSS2-ERG low sensitivity
PCA3 Progensa Prostate Cancer Antigen 3	Prostate cancer gene 3 (PCA3 or DD3) is a specific non-coding mRNA which is overexpressed in more than 95% of primary prostate tumors.	Urine-based biomarker PCA3 score over PSA, in terms of predictive value and specificity, has lower sensitivity
ConfirmMDx Hypermethylation of GSTP1, APC and RASSF1 genes, PSA	Screening patients at risk of HG PCa after an initial negative biopsy. It is clinically validated for detection of PCa in tissue from PCa-negative biopsies.	Tissue-based biomarker Tissue from prostate biopsy

**Table 4 molecules-27-05730-t004:** Common prostate cancer treatment options and potential adverse effects [88].

Treatment Option	Disease Progression	Potential Adverse Effects
Active surveillance	Localized	Illness uncertainty
Radical prostatectomy	Localized	Erectile dysfunction Urinary incontinence
External beam radiation	Localized and advanced disease	Urinary urgency and frequency, dysuria, diarrhea, and proctitis Erectile dysfunction Urinary incontinence
Brachytherapy	Localized	Urinary urgency and frequency, dysuria, diarrhea, and proctitis Erectile dysfunction Urinary incontinence
Cryotherapy	Localized	Erectile dysfunction Urinary incontinence and retention Rectal pain and fistula
Hormone therapy	Advanced	Fatigue Hot flashes and flare effect Hyperlipidemia Insulin resistance Cardiovascular disease Anemia Osteoporosis Erectile dysfunction Cognitive deficits
Chemotherapy	Advanced	Myelosuppression Hypersensitivity reaction Gastrointestinal upset Peripheral neuropathy

**Table 5 molecules-27-05730-t005:** Combination therapies for prostate cancer—completed clinical trials [116].

Primary Anticancer Agent	Secondary Anticancer Agent	Clinical Trial
Sipuleucel-T ADT Docetaxel ADT Docetaxel ADT Ipilimumab ADT ADT ADT Abiraterone Abiraterone Abiraterone	Docetaxel Radiation Thalidomide and Bevacizumab Radiation Bevacizumab Docetaxel Radiation Docetaxel Docetaxel Radiation Olaparib Radium 223 Enzalutamide	ISRCTN01534787 NCT00091364 NCT00002633/ISRCTN24991896 NCT00110214 GETUG-AFU 15 (NCT00104715) NCT00861614 CHAARTED (NCT00309985) STAMPEDE (NCT00268476) NCT00002874 NCT01972217 ERA 223 (NCT02043678)

**Table 6 molecules-27-05730-t006:** Combination therapies for prostate cancer—ongoing clinical trials [116].

Primary Anticancer Agent	Secondary Anticancer Agent	Clinical Trial	Phase and Current Status
Abiraterone Abiraterone Abiraterone ADT Apalutamide Cabazitaxel Docetaxel Olaparib	Apalutamide ADT Olaparib PROSTVAC Docetaxel, Abiraterone ADT, radiation PROSTVAC-IF Durvalumab	LACOG-0415 (NCT02867020) LATITUDE NCT03732820 NCT00450463 NCT02913196 NCT01420250 NCT02649855 NCT03810105	Phase 2, recruiting Phase 3, active and not recruiting Phase 3, recruiting Phase 2, no compiled results but completed Phase 1, recruiting Phase 1, active and not recruiting Phase 2, active and not recruiting Phase 2, recruiting Phase 2, active and not recruiting Phase 2, recruiting

**Table 7 molecules-27-05730-t007:** Anticancer drug repositioning candidates under clinical investigation for the treatment of prostate cancer [32].

Drugs	Original Use	Proposed Anticancer Mechanisms	Phase	Identifier ^∗^	Recruitment Status
Zoledronic Acid	Bisphosphonate	Inhibition of mevalonate pathway Activity of metalloproteinases	Clinical trial Phase 4	NCT00219271	Completed
Dexamethasone	Anti-inflammatory agent	Modulator of ERG activity	Clinical trial Phase 3	NCT00316927	Completed
Aspirin	Anti-inflammatory agent	COX inhibitor suppression of the neoplastic prostaglandins Inhibition of NF-κB	Clinical trial Phase 3	NCT00316927	Completed
Minocycline	Antibacterial agent	Inhibition of proinflammatory cytokines Inhibition of matrix metalloproteinases	Clinical trial Phase 3	NCT02928692	Recruiting
Celecoxib	Anti-inflammatory agent	Selective Cox-2 inhibitor Inhibition of NF-κB activity Inhibition of PDPK1/Akt signaling pathway	Clinical trial Phase 2/3	NCT00136487	Completed
Leflunomide	Immunomodulatory agent	Potent inhibitor of tyrosine kinases	Clinical trial Phase 2/3	NCT00004071	Completed
Statins	HMG-CoA reductase inhibitors	Inhibition of mevalonate pathway	Clinical trial Phase 2	NCT01992042	Completed

**Table 8 molecules-27-05730-t008:** Summary of various medicinal plants used against different types of cancers [161].

Plant Name	Phytochemical/Anticancer Agent	Type of Cancer Suppressed, Clinical and Research
*Moringa oliefera*	Niazinine A	Blood cancer (in vitro)
*Catharanthus roseus*	Vincristine and vinblastine	Testis, breast, rectum, ovary, lung, and cervical cancer (in vitro), in clinical use
*Panax ginseng*	Panaxadiol, panaxatriol	Prostate, breast, colon, ovary, lung, and colon cancer (in vitro)
*Solanum Lycopersicum*	Lycopene	Colon cancer as well as prostate (in vivo)
*Cannabis sativa*	Cannabinoid	Colorectal cancer, lung, prostate, pancreas, and breast cancer (in vitro and in vivo)
*Taxus brevifolia*	nab-Paclitaxel	Ovarian cancer as well as breast cancer (in vitro and animal studies), in clinical use
*Vitis vinifera*	Cyanidin	Colon cancer (in vitro)
*Pyrus malus*	Procyanidin, quercetin	Colon cancer (in vivo, in vitro)
*Curcuma longa*	Curcumin	Stomach cancer, prostate cancer (in vitro)
*Camellia sinensis*	Epigallocatechin gallate	Brain, bladder cancer, prostate, cervical, and bladder cancer (in vivo)
*Taxus baccata*	Cabazitaxel	Prostate cancer (in vivo), in clinical use
*Taxus baccata*	Docetaxel	Prostate, breast, and stomach cancer, in clinical use
*Taxus baccata*	Larotaxel	Pancreatic, bladder, and breast cancer (in vivo)
*Taxus brevifolia*	Paclitaxel	Breast cancer and ovarian cancer (in vivo)
*Berberis vulgaris*	Cannabisin, berberine	Liver, prostate, and breast cancer (in vivo)
*Zingiber officinale*	6-Shogaol Gingerol	Ovarian cancer (in vitro) Ovarian, colon, and breast cancer (both in animal experiments and in vitro experiments)
*Aloe vera*	Alexin B, emodin	Stomach cancer and leukemia (in vivo)
*Vaccinium macrocarpon*	Hydroxycinnamoyl ursolic acid	Prostate and cervical cancer (in vitro)
*Hibiscus mutabilis*	Lectin	Breast and liver cancer (in vitro)
*Momordica charantia*	Cucurbitane-triterpene, charantin	Breast and colon cancer (in vitro)
*Podophyllum peltate*	Etoposide Teniposide	Lung, testicular, leukemia, lymphoma Hodgkin’s lymphoma
*Curcuma longa*	Curcumin	Stomach cancer (in vitro) Lung, prostate, skin, colon breast, lung, colon, prostate, liver esophagus (in vitro)
*Cicer arietinum Crocus*	Bowman–Birk-type protease	Prostate as well as breast cancer (in vitro)

## Data Availability

Information was available in the public domain. Databases have been provided in Section 2.

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
