# Peer review of "Prostate Cancer Review: Genetics, Diagnosis, Treatment Options, and Alternative Approaches"

_molecules, 2022, doi:10.3390/molecules27175730_

Round 1

Reviewer 1 Report

In the present narrative review, the authors report on genetics, diagnosis, treatment options and alternative approaches in patients diagnosed with prostate cancer.

Overall, attention has been paid to the accuracy of the text and the article provides an overview of the research area.

Particularly, I would suggest to include a M&M section indicating if a review protocol exists, describing all information sources in the search, date last searched, process for selecting studies (i.e., screening, eligibility, etc.) as well as provide some information on method of data extraction.

For example, they could provide all this information with a flow diagram.

Author Response

Materials and Methods section describing all information sources in the search, process for selecting studies providing information on the method of data extraction was added

Reviewer 2 Report

This review manuscript is generally well written. I have only minor comments.
1. The authors describes hormone therapy(1.5.6; line 322 -) , but there is no explanation regarding flutamide
and chlormadinone. It might be better move 1.5.8.1 abieaterone to after hormone therapy?
2. Figs. 1 and 2 is too small to read it.
3. Inconsistent character style in small titles; There is no uniformity as to whether the first letter should be capitalized or
lowercase.

Author Response

Flutamide and chlormadinone were added under hormonal therapy. Paragraph 1.5.8.1 abieaterone was moved to after hormone therapy as requested. Figures 1 and 2 were enlarged and the document was reviewed for
inconsistent character style in small titles as requested. 

Reviewer 3 Report

Reviewer’s comments to the author:

1.    In treatment challenges, what are the important factors or genetic mutations or alternations that can lead to the growth, metastasis, tumorigenesis, tumor microenvironment, and tumor environment interaction of prostate cancer?

2.   What is the role of next-generation sequencing (NGS) in the ongoing research on precision medicine, anticancer drug development, and solving drug resistance problems for prostate cancer?

3.   The author adds a concise description of the important learning objectives of the review article in the conclusion or introduction.

Author Response

The important factors or genetic mutations or alternations that can lead to the growth, metastasis, tumorigenesis, tumor microenvironment, and tumor environment interaction of prostate cancer were added. The role of next-generation sequencing (NGS) in the ongoing research on precision medicine, anticancer drug development, and solving drug resistance problems for prostate cancer was also added as requested.

Lastly, a concise description of the important learning objectives of the review article in the introduction was added as requested by the reviewer.